# Role and Mechanism of Cold Plasma in Inactivating *Alicyclobacillus acidoterrestris* in Apple Juice

**DOI:** 10.3390/foods12071531

**Published:** 2023-04-04

**Authors:** Hao Ding, Tiecheng Wang, Yuhan Sun, Yuxiang Zhang, Jianping Wei, Rui Cai, Chunfeng Guo, Yahong Yuan, Tianli Yue

**Affiliations:** 1College of Food Science and Engineering, Northwest A&F University, Yangling 712100, China; dheron@163.com (H.D.); sunyh@nwafu.edu.cn (Y.S.); cairui@nwsuaf.edu.cn (R.C.); gcf@nwafu.edu.cn (C.G.); 2Laboratory of Quality & Safety Risk Assessment for Agro-Products (Yangling), Ministry of Agriculture, Yangling 712100, China; 3College of Natural Resources and Environment, Northwest A&F University, Yangling 712100, China; wangtiecheng2008@126.com; 4Key Laboratory of Plant Nutrition and the Agri-Environment in Northwest China, Ministry of Agriculture, Yangling 712100, China; 5College of Food Science and Technology, Northwest University, Xi’an 710069, China; zhangyx729@nwu.edu.cn (Y.Z.); jianpingwei0327@nwu.edu.cn (J.W.)

**Keywords:** *Alicyclobacillus acidoterrestris*, cold plasma, inactivation, mechanism, ROS, singlet oxygen, apple juice, juice quality

## Abstract

*A. acidoterrestris* has been identified as the target bacterium in fruit juice production due to its high resistance to standard heat treatment. Multiple studies have shown that cold plasma can effectively inactivate pathogenic and spoilage microorganisms in juices. However, we are aware of only a few studies that have used cold plasma to inactivate *A. acidoterrestris.* In this study, the inactivation efficacy of cold plasma was determined using the plate count method and described using a biphasic model. The effects of the food matrix, input power, gas flow rate, and treatment time on inactivation efficacy were also discovered. Scavenging experiments with reactive oxygen species (•OH, •O_2_^−^, and ^1^O_2_), scanning electron microscopy (SEM), Raman spectra, as well as an in vitro toxicology assay kit, were used to determine the inactivation mechanism. According to the plate count method, a maximum reduction of 4.14 log CFU/ mL could be achieved within 7 s, and complete inactivation could be achieved within 240 s. The scavenging experiments showed that directly cold plasma-produced singlet oxygen plays the most crucial role in inactivation, which was also confirmed by the fluorescence probe SOSG. The scanning electron microscopy (SEM) and Raman spectra showed that the cold plasma treatment damaged the membrane integrity, DNA, proteins, lipids, and carbohydrates of *A. acidoterrestris*. The plate count results and the apple juice quality evaluation showed that the cold plasma treatment (1.32 kV) could inactivate 99% of *A. acidoterrestris* within 60 s, with no significant changes happening in apple juice quality, except for slight changes in the polyphenol content and color value.

## 1. Introduction

*Alicyclobacillus* is a target bacterium in the design of pasteurization processes for acidic fruit-based foods and beverages. Since 1984, fruit juice contamination brought on by *Alicyclobacillus* has been widely reported in Europe, America, Brazil, Argentina, and some other places, bringing fruit juice producers significant financial losses as well as leading to a loss of consumer confidence [1,2]. According to the American Food Processors Association (NFPA) research, *Alicyclobacillus* is responsible for 35% of fruit juice pollution, with *Alicyclobacillus acidoterrestris* being the primary culprit [3]. *A. acidoterrestris* is a rod-shaped, Gram-positive, aerobic spoilage bacterium. It is capable of growing in a wide temperature range of 25–60 °C with an optimum between 40 and 45 °C, as well as in a pH range of 2.0–7.0 with an optimum between 3.5 and 4.0. The high content of ω-alicyclic fatty acids in the cell membrane of *A. acidoterrestris* causes its high heat and acid resistance. *A. acidoterrestris* can survive as spores during commercially applied pasteurization process (88 to 96 °C for 2 min, or 90 to 95 °C for 30 to 60 s), and it then germinates and grows in juices, causing off-flavors and sediment [1]. Furthermore, traditional inactivation conditions can negatively impact juice quality. Therefore, novel, non-thermal inactivation methods are needed to improve food preservation and quality. Non-thermal techniques, such as high hydrostatic pressure (HPP), ultrasound, ohmic heating, microwave, UV-C light, as well as essential oils and plant extracts, have been widely studied to inactivate *A. acidoterrestris* in juices [2,4]. However, to the best of our knowledge, very few studies have reported the inactivation efficacy and mechanism of cold plasma on *A. acidoterrestris*.

Cold plasma is considered an efficient, low-temperature, and environmentally friendly inactivation method, and it has been widely studied in the fields of medicine, food, agriculture, and environmental protection. Studies have shown that cold plasma can efficiently sterilize bacteria, yeast, spores, and viruses in liquid foods, such as juices and milk [5,6,7]. Research on inactivating *Escherichia coli* [5], *Zygosaccharomyces rouxii* [8], and *Citrobacter freundii* [9], reported at least a 4-log reduction in apple juice. However, the typical treatment volume of normal DBD and plasma jet is very small, and more research with large sample sizes is required to predict industrially important processing parameters, as suggested by E. Ozen et al. [6]. Thus, the typical surface discharge plasma with a volume of 330 mL was used in this study to inactivate *A. acidoterrestris*.

Cold plasma is considered the fourth state of matter, consisting of charged particles (H_3_O^+^, H^+^, and OH^−^), reactive oxygen species (ROS, such as ^1^O_2_, •O_2_^−^, and •OH, O_3_), reactive nitrogen species (RNS, such as NO_3_^−^, NO_2_^−^, and ONOOH), excited molecules, and UV photons and electrons [10]. Typically, ROS and RNS are regarded as the primary causes of bacteria inactivation, which could attack bacteria and cause cell wall damage by electroporation and lipid oxidation, leading to DNA damage and protein modulation [11]. In recent years, more and more research has reported that •OH, ^1^O_2_, •O_2_^−^, and ONOOH, but especially ^1^O_2_, play important roles [12,13,14,15,16]. Our previous study using surface discharge plasma has also reported that •OH, ^1^O_2_, and •O_2_^−^ play important roles in inactivating *A. spiroides* and *Escherichia coli* [17,18]. However, which species plays the most important role and the production pathway is still controversial.

Thus, to find a new way for controlling *A. acidoterrestris*, the goal of this study was to investigate the inactivation efficacy and mechanism of cold plasma on *A. acidoterrestris*, as well as to determine the effect of cold plasma treatment on apple juice quality. Scavenging experiments were used to elucidate the inactivation mechanism and clarify which species is the most important inactivation factor. Scanning electron microscopy (SEM) was employed to elucidate the morphological changes in cells upon cold plasma treatment. Raman spectra were recorded to detect damage to cell components, and an in vitro toxicology assay kit (resazurin-based) was used to measure metabolic capacity. The main quality parameters of apple juice were determined, including pH, total sugar, reducing sugar, total acid, total polyphenols, and color.

## 2. Materials and Methods

### 2.1. Cold Atmospheric Plasma System

The cold atmospheric plasma device mainly consists of a power supply, a discharge reactor, and a gas pump, as shown in Figure 1. The power supply was provided by the Dalian University of Technology, China, with the frequency set to 7.0 kHz and the discharge voltage set between 0 and 30 kV. The discharge reactor is the same as described by Wang et al. [19], which consists of a glass vessel (30 cm long and 1 mm in thickness) with an iron spring (18 cm long and 1 cm in diameter). Inside the vessel, the iron spring is connected to the power supply as a high-voltage electrode, and an aerator is placed at the bottom of the vessel. The total treatment volume was 330 mL in this study. Air was supplied by a gas pump, dried using allochromatic silica gel, and then pumped into the discharge reactor with a variable gas flow rate of 0–1000 mL/min. When the power supply was switched on, dry air was ionized by high-voltage electrons in the reactor. Reactive species were produced, carried into the liquid media by the gas flow, and came into contact with the liquid.

### 2.2. Strains and Growth Condition

Strains of *A. acidoterrestris* (DSM 3922, DSM 3923, and DSM 2498) were obtained from the German Collection of Microorganisms and Cell Cultures (DSMZ, Braunschweig, Germany). All strains were preserved at −80 °C in an *A. acidocaldarius* medium (AAM) with 30% glycerol before use. The AAM broth contained yeast extract (2.00 g), glucose (2.00 g), (NH_4_)_2_SO_4_ (0.40 g), MgSO_4•_7H_2_O (1.00 g), CaCl_2_ (0.38 g), and KH_2_PO_4_ (1.20 g) in distilled water (1 L). The pH was adjusted to 4.0 with 1 M H_2_SO_4_. The vegetative cells were cultivated in the AAM broth at 45 °C with shaking at 120 rpm for 14 h and then washed three times with 0.85% saline before inoculation [20].

### 2.3. Cold Plasma Treatment

A cocktail of vegetative *A. acidoterrestris* cells (DSM 3922, DSM 3923, and DSM 2498) was inoculated into 0.85% saline and 12% apple juice, respectively, at a concentration of 1.0 × 10^7^ CFU/ mL. When the matrix was 0.85% saline, the gas flow rate was set to 130 mL/min, and the cold plasma input power was adjusted to 1.32, 2.20, and 4.64 kV, respectively; then, the input power was set to 6.86 kV, and the gas flow rate was adjusted to 80, 130, and 180 mL/min, respectively. When the matrix was 12% apple juice, the gas flow rate was set to 130 mL/min, and the cold plasma input power was adjusted to be at 1.32, 4.64, and 6.86 kV, respectively; then, the input power was set to 6.86 kV, and the gas flow rate was adjusted to 130, 180, and 230 mL/min, respectively. The sampling time was 0, 7, 15, 30, 60, 120, 240, 360, 480, and 720 s. All experiments were carried out in triplicate.

### 2.4. Bacterial Recovery and Enumeration

After the cold plasma treatment, 100 μL of the sample was taken, and gradient dilution was performed. Then, a plate count experiment was conducted for *A. acidoterrestris* on the AAM for 36 h at 45 °C [21]. Each experiment was repeated at least three times.

### 2.5. Modeling of Inactivation Data

GInaFiT (Geeraerd and Van Impe Inactivation Model Fitting Tool), a freeware add-in for Microsoft^®^ Excel, was used to model the inactivation kinetics that was discovered [22]. GInaFiT offers 10 common kinetic models to simulate inactivation curves, and a biphasic model was chosen to describe the inactivation here [23]. Geeraerd et al. [22] described the model used as follows:

Biphasic model:(1)$$logN=logN_{0}+log(f\times exp^{-k_{max1}t}+\left(1-f\right)\times exp^{-k_{max2}t})$$
where *f* is the proportion of the initial population in a major population; (1 − *f*) is the fraction of the initial population in a minor population; and *k_max_*_1_ and *k_max_*_2_ are the specific inactivation rates of the two populations. The adj-R^2^ and root mean square error (RMSE) values were chosen as the “goodness-of-fit” indicators for the evaluation models.

### 2.6. Scavenging Experiments

In these experiments, 1 M L-histidine, 500 U SOD, and 1 M D-mannitol were used for scavenging singlet oxygen (^1^O_2_), superoxide anion (•O_2_^−^), and hydroxyl radical (•OH), respectively [13,14]. The initial concentration of the vegetative cells of *A. acidoterrestris* (DSM 3922, DSM 3923, and DSM 2498) was set at 1.0 × 10^7^ CFU/mL and then treated by cold plasma with a voltage of 1.32 kV, a gas flow rate of 130 mL/min, and a treatment time of 0, 15, 30, 60, and 120 s, respectively. A plate count experiment was used for sterilization efficacy determination.

### 2.7. Singlet Oxygen Determination

Singlet Oxygen Sensor Green^®^ (SOSG, Kit-S36002, Thermo Fisher, USA) was diluted to 10 µM and used for detecting singlet oxygen produced with and without 1 M L-histidine, with a treatment time of 0, 2, 6, 10, and 15 s, respectively, a voltage of 1.32 kV, and a gas flow rate of 130 mL/min. The product of SOSG that reacted with singlet oxygen was detected by a Biotek Synergy HT fluorescence plate reader (BioTek, Winooski, VT, USA), using an excitation wavelength at 485 nm and an emission wavelength at 528 nm [14].

### 2.8. SEM Analysis

The cell surface integrity of *A. acidoterrestris* was examined via SEM [24]. A concentration of 1.0 × 10^7^ CFU/mL vegetative cells was inoculated and treated at 130 mL/min for 0, 15, 30, and 60 s, respectively, by cold plasma. Subsequently, the cells were collected by centrifugation at 5000 rpm for 10 min. The vegetative cells were washed in PBS (pH 7.3) three times, inoculated on coverslips (7 mm × 7 mm), and fixed with 2.5% glutaraldehyde overnight at 4 °C. After washing with PBS three times, the vegetative cells on the coverslips were transferred to 4% isoamyl acetate for 2 h. Liquid CO_2_ was used to dehydrate the cells, and they were sputter coated with approximately 30 nm of gold by Leika 1045 (Hitachi, Tokyo, Japan), using argon gas as the ionizing plasma. The images were obtained with an S-570 SEM (Hitachi, Tokyo, Japan) with secondary electrons at an acceleration voltage of 20 kV at room temperature.

### 2.9. Metabolic Capacity

An in vitro resazurin-based toxicology assay kit (Tox-8, Tox-8, Sigma-Aldrich, St. Louis, MO, US) was applied to measure the metabolic capacity of *A. acidoterrestris* treated by cold plasma (1.32 kV, 130 mL/min, and 60 s) [25]. The resazurin fluorescence probe solution (100 μL) was mixed with a suspension of plasma-treated bacteria (1 mL) and incubated in the dark for 2 h. Bacteria with metabolic capacity can convert resazurin (blue and weakly fluorescent) to resorufin (pink and highly fluorescent). The suspension was diluted 20 times before analysis by a spectrofluorometer (Fluoromax-4, JOBIN YVON Technology, Horiba Jobin Yvon Inc., Edison, NJ, USA). The excitation/emission (Ex/Em) wavelengths were 560/590 nm. The calculation of the ratio of plasma-treated bacteria with metabolic capacity was based on setting untreated ones as 100% [25].

### 2.10. Raman Spectrum Analysis

A Renishaw Raman spectrometer system (inVia System; Renishaw plc, Gloucestershire, UK), equipped with a Leica microscope and a 532 nm laser source (50 mW), was used to acquire the Raman spectra of *A. acidoterrestris* treated for 0, 60, and 360 s by cold plasma with the voltage set at 1.32 kV and the gas flow rate set at 130 mL/min. The grating scan type, laser power, and exposure time were set to “extended”-“step”, 10%, and 10 s, respectively [26]. The microscope objective was 100×, and the detection range was from 600 to 1800 cm^−1^. To analyze the data, smoothing and polynomial subtraction were used for eliminating instrumental noises and baseline offsets by WiRE 5.0. Principal component analysis (PCA) was performed using Origin 2021 to determine the similarity and differences among the samples.

### 2.11. Evaluation of Apple Juice Quality

According to Wei et al. [27], the soluble solid content (SSC) was detected by a hand-held refractometer (WYT-4; Quanzhou Optical Instrument Co., Ltd., Quanzhou, China). The pH values were measured by a pH meter (PHS3C, Shanghai INESA Scientific Instrument Co., Ltd., Shanghai, China). The total acid content was determined with 0.05 mol/L NaOH and was expressed in terms of tartaric acid. Fehling’s reagent method was used to determine the total sugar content. The color of apple juice was assessed using the values of L* (brightness/darkness), a* (redness/greenness), and b* (blueness/yellowness), which were measured via a colorimeter (X-Rite Ci7600, Grand Rapids, MI, America). The total color difference was calculated as follows:ΔE = (ΔL*^2^ + Δa*^2^ + Δb*^2^)^1/2^(2)

The total color difference (ΔE) was classified as “unnoticeable” (0.0 to 0.2), “very little” (0.2 to 0.5), “small” (0.5 to 1.5), “distinct” (1.5 to 3.0), “very distinct” (3.0 to 6.0), “great” (6.0 to 12.0), and “very great” (>12.0).

### 2.12. Statistical Analysis

All experiments were carried out at least in triplicate. Statistical analysis was performed with Origin 7.0. The means and standard deviations were calculated, and analysis of variance (ANOVA) and Tukey’s test were applied. A probability level of less than 0.05 (*p* < 0.05) was considered statistically significant.

## 3. Results

### 3.1. Inactivation Efficacy of A. acidoterrestris by Cold Plasma

The antibacterial efficacy of cold plasma on *A. acidoterrestris* in saline and apple juice, using different voltages and gas flow rates, is shown in Figure 2. When the matrix is 0.85% saline, the bactericidal rate can reach 99% within 7 s, and the bactericidal efficiency increased with increasing voltage (Figure 2A). However, after prolonged treatment time, the effect of the treatment has plateaued. Increasing the flow rate improves inactivation after 7 s at 6.86 kV and shortens the time required to reach total mortality, and the fastest fully sterile state can be achieved in 240 s with a gas flow rate of 180 mL/min (Figure 2B).

As shown in Figure 2C,D, a bactericidal rate of 99% is achieved within 120 s in apple juice, and the bactericidal effect increases with increasing voltage (1.32–6.86 kV). However, after 120 s, the effect has plateaued for all voltage treatments, and the bactericidal rate at 6.86 kV could not be improved by increasing the gas flow (Figure 2D).

Overall, higher voltage, higher gas flow rate, and longer treatment time of cold plasma resulted in better bactericidal effect on *A. acidoterrestris* in saline. However, in apple juice, the bactericidal rate slowed or plateaued after 120 s of treatment, and the bactericidal rate did not improve by increasing the atmospheric flow rate and treatment time.

### 3.2. Kinetic Models

According to the bactericidal curves of *A. acidoterrestris* treated by cold plasma under different voltages in apple juice (Figure 2C), the biphasic model was used for data fitting and evaluation (Table 1). The *f* value represents the number of bacteria sensitive to the treatment. The *f* value is 0.99 or 1.00, indicating *A. acidoterrestris* is very sensitive to the cold plasma treatment. *K_max_*_1_ and *k_max_*_2_ represent the germicidal rate of the first and second sections, respectively. For different voltages, *k_max_*_1_ is significantly higher than *k_ma_*_x2_, and when the voltage is 1.32, 4.64, and 6.86 kV, the value of *k_max_*_1_ is 0.07, 0.20, and 0.30, respectively. Thus, *k_max_*_1_ increases significantly with increasing voltage, indicating that the voltage increase also increases the bactericidal rate. According to adj-R^2^ (all >0.95) and RMSE, the biphasic model fits the survival curves of *A. acidoterrestris* treated by cold plasma well.

### 3.3. Inactivation Mechanism of Cold Plasma Used on A. acidoterrestris

The investigation showed that 1.32 kV treatment already resulted in a good reduction in both saline and apple juice; thus, the mechanism was determined with the voltage set at 1.32 kV and the gas flow rate set at 130 mL/min.

#### 3.3.1. Scavenging Experiments

To elucidate which species has the most important bactericidal effect, different scavengers (1 M L-histidine, 500 U SOD, and 1 M D-mannitol) were used for scavenging singlet oxygen (^1^O_2_), superoxide anion (•O_2_^−^), and hydroxyl radical (•OH), respectively (Figure 3). After adding 1 M L-histidine, no reduction (*p* > 0.05) was found, as shown in Figure 3A, indicating that all *A. acidoterrestris* were protected from being inactivated when scavenging ^1^O_2_. No obvious difference (*p* > 0.05) was shown at 15 s after adding 500 U SOD, indicating •O_2_^−^ did not play the most important role. After adding 1 M D-mannitol, a 1.29-log reduction happened within 15 s, compared to the control with a 3.21-log reduction within 15 s (Figure 3C), indicating that approximately 99% *A. acidoterrestris* still were inactivated. Thus, •OH does not play the most important role in inactivating *A. acidoterrestris*.

Although L-histidine and D-mannitol are regarded as scavengers for ^1^O_2_ and •OH, respectively, L-histidine can also scavenge •OH, and D-mannitol can scavenge little ^1^O_2_ as well [14]. Thus, the fluorescence probe SOSG was used to identify the production of ^1^O_2_ with and without L-histidine, and the results showed that ^1^O_2_ could be rapidly produced at 1.32 kV in the control, and the intensity was significantly inhibited after adding L-histidine (Figure 3D). Thus, we can conclude that singlet oxygen plays the most important role in inactivating the vegetative cells of *A. acidoterrestris* using cold plasma.

#### 3.3.2. Morphological Changes

Membrane damage caused by cold plasma in *A. acidoterrestris* is shown in Figure 4. In the SEM images, the damage intensity of vegetative cells increases with increasing treatment time (0, 15, 30, 60 s). Initially, the membranes appear smooth and intact (Figure 4A) but quickly develop small holes and wrinkles (Figure 4B,C). Finally, marked deformation and cell rupture are observed (Figure 4D). Therefore, cold plasma causes irreversible and severe membrane damage to *A. acidoterrestris* cells, and the cell contents escape from the ruptured cells.

#### 3.3.3. Raman Spectra

Figure 5 shows the Raman spectra and PCA plot of *A. acidoterrestris* treated with cold plasma at 0, 60, and 360 s, respectively. The predominant Raman peaks are between 600 and 1800 cm^−1^ and are labeled in Figure 5A. The peaks can be categorized as proteins (bands around 853, 1005, 1252, and 1665 cm^−1^), nucleic acids (bands around 726, 783, 936, 1101, 1340, and 1577 cm^−1^), lipids (bands around 1128 and 1452 cm^−1^), and carbohydrates (bands around 1035 cm^−1^) [25]. Clear segregation is shown between the data clusters of 0 s and 360 s, but not between 0 s and 60 s or between 60 s and 360 s (Figure 5B). The first two PC scores explain 74.4% of the variances in the spectral data, with all peaks showing a positive correlation with PC 1 (54.2%), except the bands around 853 cm^−1^, which have a positive correlation with PC 2 (20.2%).

To better understand the targeting point and inactivation mechanism, PCA analyses were conducted individually for lipids (Figure 5C), proteins (Figure 5D), and DNA (Figure 5E). Clear segregation is shown between the data clusters of 0 s and 60/360 s, indicating the cold plasma treatment causes C-H_2_ deformation (1453 cm^−1^) and C-N stretching (1128 cm^−1^) of lipids (Figure 5C). The changes in amide I (1665 cm^−1^), amide III (1252 cm^−1^), and phenylalanine (1005 cm^−1^) cause a segregation of the data clusters (PC1 (76.1%)) between 0 s and 360 s (Figure 5D), while the changes in nucleic acid (A: 726 cm^−1^; C/T: 783 cm^−1^; A/G: 1577 cm^−1^) cause a clear segregation of the data clusters (PC2 (21.7%)) between 0/60 s and 360 s (Figure 5E). C-C deformation of carbohydrates (1035 cm^−1^) was also found to have a positive correlation with PC1 (54.2%) which separates the data clusters of 0 s and 360 s (Figure 5B). Thus, we can conclude that the cold plasma treatment causes statistically significant changes in the cellular contents (lipids, proteins, DNA, and carbohydrates,) of *A. acidoterrestris*, and the changes become more drastic with a longer treatment time.

#### 3.3.4. Metabolic Capacity

The percentage of metabolic capacity of *A. acidoterrestris* vegetative cells treated with cold plasma with and without L-histidine is shown in Figure 6. The results show that 33% of *A. acidoterrestris* still have metabolic capacity after 60 s treatment with cold plasma, although the plate count results indicate that only 0.01% of the bacteria have culturability (Figure 2). When microorganisms fail to grow on the medium while preserving some of their metabolic activity, this is known as a nonculturable state (VBNC) [28]. Thus, approximately 33% of *A. acidoterrestris* reach a VBNC state after the cold plasma treatment for 60 s. However, when adding L-histidine, the plate count results show that the survival rate is 100%, and the metabolic capacity is even greater than 100%. No death or metabolic damage happened when L-histidine was added into the system, indicating that L-histidine successfully protects *A. acidoterrestris* from being attacked by singlet oxygen produced by cold plasma, and validating the previous findings in Figure 3.

#### 3.3.5. The Effects of Cold Plasma Treatment on Apple Juice Quality

To clarify the application’s feasibility, cold plasma’s impact on apple juice quality was measured with a voltage of 1.32, 4.64, and 6.86 kV at a treatment time of 0, 30, 60, and 120 s, respectively, and with a gas flow rate of 130 mL/min.

#### 3.3.6. Changes in the Quality Parameters of Apple Juice

As shown in Table 2, overall, the cold plasma treatment with different applied voltages (1.32, 4.64, and 6.86 kV) and treatment times (0, 30, 60, and 120 s) does not significantly affect total soluble solids, pH, and total acid of apple juice (*p* > 0.05). When the voltage was set at 1.32 kV, no significant changes happened in the main quality of apple juice, except for a decrease in polyphenol content (*p* < 0.05). However, when the voltage increased to 4.64 and 6.86 kV, the sugar content fluctuated, and the total phenol content significantly decreased (*p* < 0.05). These results show that the higher the voltage and the longer the treatment time of the cold plasma treatment, the severer the damage to the total phenol content of apple juice.

#### 3.3.7. Changes in the Color Value of Apple Juice

The apple juice treated by cold plasma was sampled with different input powers and treatment times, and the parameters, L*, a*, and b*, of the three-color coordination system were measured (Figure 7). The L* value of apple juice significantly increases (*p* < 0.05) with treatment time, indicating the cold plasma treatment increases the brightness of apple juice. Although there are fluctuations in the value of a*, the overall trend is increasing, indicating the apple juice treated by cold plasma becomes greener. In addition, the b* values decrease with increasing voltage and treatment time, indicating the yellow intensity of the apple juice reduces with the extension of treatment time. After 30 s of treatment, the value of ΔE*ab, which represents the color difference, ranges from 2 to 13. It is also revealed that the cold plasma treatment has a significant impact (*p* < 0.05) on the color of apple juice.

## 4. Discussion

Here, the effect of cold plasma on inactivating *A. acidoterrestris* was studied. The results showed that cold plasma could effectively inactivate *A. acidoterrestris*, and the inactivation efficacy was significantly influenced by the power supply, flow rate, and treatment time. Singlet oxygen plays the most important role in inactivating *A. acidoterrestris* by cold plasma. Substantial destruction of cell membranes was observed, and damage on the surface, as well as to cellular components, such as lipids, carbohydrates, proteins, and DNA, occurred. In apple juice, a low-voltage (1.32 kV) treatment had no significant effects on the whole quality of apple juice, except for a slight change in polyphenol content and color value. However, with an increase in voltage and treatment time, the cold plasma treatment significantly decreased the total polyphenol content and color value and resulted in a fluctuation in the content of total sugar.

### 4.1. Inactivation Efficacy of Cold Plasma

Overall, the survival curves show that higher input power, higher gas flow rate, and longer treatment time result in better inactivation (Figure 2). Deng et al. [29] also reported the application of 16, 20, and 25 kV resulted in 1.0-, 2.43-, and 4.12-log reductions, respectively, of *Escherichia coli* (*E. coli*) using dielectric barrier discharge plasma. Han et al. [30] reported a decrease in *E. coli* survival from 2 logs to complete inactivation when the treatment time increased from 1 to 3 min. Wang et al. [31] reported that the D value of yeast treated by cold plasma in apple juice decreased from 4.85 min to 2.47 min with the voltage increasing from 15 kV to 21 kV. Higher input power indicates higher input energy intensity, which corresponds to the generation of a greater amount of reactive species. Therefore, higher inactivation efficacy can be achieved [11].

Better inactivation of *A. acidoterrestris* was achieved with a higher gas flow rate (Figure 2B). The plasma apparatus in this study can pump gas containing reactive species as bubbles into liquid samples. Within a certain range, the higher the gas flow rate, the higher the possibility for the reactive species in the bubbles to contact with bacteria, thus resulting in better inactivation [32,33]. However, when using apple juice as the matrix, a tail occurred in which at least 2-log *A. acidoterrestris* continued to survive after being treated for 2 to 12 min. Additionally, increasing the gas flow rate cannot increase the inactivation, as shown in Figure 2D. This might be because the components in apple juice, such as carbohydrates, proteins, and polyphenols, can either block or interact with reactive species.

Other studies have also reported that gas types, exposure mode, and matrix composition can influence the inactivation efficacy [14,30,34]. Thus, these parameters should be optimized to increase the contact possibility and contact time to efficiently inactivate *A. acidoterrestris* and lower the cost of treatment.

### 4.2. Singlet Oxygen Plays the Most Important Role in Inactivating A. acidoterrestris by Cold Plasma

ROS (O_3_, •O_2_^−^, H_2_O_2_, •OH, and ^1^O_2_) and RNS (NO, NO_2_^−^, NO_3_^−^, and ONOOH) are normally regarded as the main reason for cold plasma sterilization [10]. However, which one plays the most important role is still controversial, and it differs with the reactor type, gas type, etc. Recent studies have shown results that regard ^1^O_2_ as having the most important role in inactivating *Staphylococcus aureus* [13,15], bacteriophage T4 [16], yeast [12], and virus [14]. Such studies suggest that as long as researchers detect singlet oxygen, they will find that singlet oxygen plays the most important role. Why does singlet oxygen have the most important role?

Singlet oxygen is the lowest excited state of oxygen, and it is easier to be activated with a redox potential of 0.98 eV than any other ROS [35,36]. Singlet oxygen has a long lifetime, lasting from tens of milliseconds to 70 min in air depending on the types and concentrations of other species, 3 µs in water, and 100 ns–3 µs in cells, thereby allowing it to transfer and react with distant targets [36,37]. Because of its small size, singlet oxygen can diffuse easily through cellular structures and polymers and then react with inner cellular macromolecules. Additionally, the unique electronic structure and excess energy make it highly reactive against macromolecules with high electron-contained double bonds, such as proteins, DNA, and lipid acids [16].

However, the production pathway of singlet oxygen has not been clearly reported. There are five main ways for singlet oxygen production [13,38,39,40,41]: (1) direct energy transfer to oxygen (Equations (3) and (4)); (2) second-order reaction originated from a single atom of oxygen (Equation (5)); (3) second-order reaction originated from chlorine ions (Equations (6)–(8)); (4) second-order reaction originated from hydroxyl and superoxide anion (Equations (9)–(12)); and (5) second-order reaction originated from RNS (Equation (13)).
(3)$$O_{2}+e\to O_{2}\left(a{\;}^{1}{∆}_{g}\right)+e$$
(4)$$O_{2}\left(a{\;}^{1}{∆}_{g}\right)+e\to O_{2}\left(b^{1\overset{+}{\underset{g}{\sum }}}+e\right)$$
(5)2O→O12
(6)$$2{\mathrm{Cl}}_{2}{}^{-}\to {\mathrm{Cl}}_{2}+2{\;\mathrm{Cl}}^{-}$$
(7)$${\mathrm{Cl}}_{2}+{\;H}_{2}O\to \mathrm{HOCl}\;+{\;\mathrm{Cl}}^{-}+H^{+}$$
(8)OCl−+ H2O2→H2O + Cl−+O12
(9)O2−+H2O2→O12+ OH−+OH
(10)O2−+OH→O12+ OH−
(11)O2−+HO2→O12+ OH2−
(12)HO2+ HO2 →O12+ H2O2
(13)O3+ NO2− →O12+ NO3−

The scavenging experiments indirectly demonstrate that directly cold plasma-produced singlet oxygen is the major way for singlet oxygen supplement within 15 s (Figure 3). There is no difference between the sterilization effect of vegetative cells in water and in saline, indicating that the pathway related to chlorine ions maybe not the main one in the second-order singlet oxygen production. However, adding enzyme SOD did not inhibit the reduction in vegetative cells within 15 s, but no reduction happened within 15–60 s, indicating the superoxide anion may participate in the second-order singlet oxygen production but not in the major pathway (Figure 3B). The fourth pathway related to peroxynitrite may also play little role because the contents of nitrite and nitrate are pretty low within 15 s at 1.32 kV. Thus, we could conclude that directly cold plasma-produced singlet oxygen (Equations (3)–(5)) is the main reason for *A. acidoterrestris* inactivation.

### 4.3. The Influence of Cold Plasma on A. acidoterrestris

The cell membrane damage of *A. acidoterrestris* by cold plasma is shown in Figure 4. Polysaccharides, lipids, proteins, and DNA on cell wall, as well as other cellular components, are also damaged by the cold plasma treatment (Figure 5). Reactive oxygen species (ROS) and reactive nitrogen species (RNS) are normally regarded as the most important bactericidal species, which can attack on cell wall and react with cellular components. Severe damage to the cell wall and inner components (such as polysaccharides, lipids, DNA, and proteins) was also reported in *Staphylococcus aureus* (*S. aureus*), *E. coli*, *MS2* virus, yeast, *Psychrobacter glacincola*, *Listeria monocytogenes*, and *Salmonella* Enteritidis treated by cold plasma [24,30,42,43,44,45,46].

As shown in Figure 6, *A. acidoterrestris* exhibits a VBNC state after cold plasma treatment, and similar results have been reported in other studies [47]. As reported by Xu et al. [25], when bacteria (>99.999%) lost their culturability, 14% of *S. aureus* and 0.5% of *E. coli* retained metabolic capacity after 20 min of plasma treatment. The biggest problem posed by VBNC microorganisms is that they cannot be identified using the standard plate count method, but they may still be virulent and resilient to various conditions. Therefore, the presence of VBNC bacteria during the processing of food could pose a serious risk to public health and food safety. Thus, preventing bacteria from entering the VBNC state and achieving complete mortality is necessary to ensure food safety. Liao et al. [28] reported that oxidative stress caused by reactive species is the main reason for inducing *S. aureus* to a VBNC state. Although increasing the treatment time can increase cell damage, this results in higher costs and greater impacts on juice quality [48]. More research on this tendency should be conducted in the future.

### 4.4. Changes in Apple Juice Quality

In apple juice, a low-voltage (1.32 kV) treatment has no significant effects on the main quality parameters of apple juice, except for changes in polyphenol content and color value (Table 2 and Figure 7). However, with an increase in voltage and treatment time, the content of total sugar fluctuates, and the total polyphenol content significantly decreases while the color value significantly changes (*p* < 0.05). Similar results were reported by Liao et al. [5], who showed that a dielectric barrier discharge (DBD)-ACP has slight effects on the pH, TSS, TA, color values, and total phenolic content of apple juice. However, significant changes in pH, TA, color, and total phenolic content are observed in apple juice at a higher input power. Thus, high voltage and long treatment time will seriously affect the sugar–acid ratio, which then affects the taste and flavor of apple juice.

For the changes in the color values of apple juice, L* increases and b* significantly decreases (*p* < 0.05) after the cold plasma treatment. Although there are fluctuations in the data of a*, the overall trend is increasing (*p* < 0.05). This means the cold plasma treatment has resulted in brighter and greener apple juice. Interestingly, Wang et al. [49] reported apple juice became brighter and yellower after cold plasma treatment, and Liao et al. [5] reported the L* value declined, while the a* and b* values increased. Umair et al. found that high-voltage electric field cold plasma (HVCP) resulted in an obvious increase in all color indices (L*, a*, and b*) of fresh carrot juice [50]. Wang et al. found that cold plasma treatment caused an increase in L* and b* and a decrease in a* in jujube juice [51]. The discrepancies in these findings may be caused by different compositions and existence of various pigments in different juice types. Reactive species can react with these pigments and vitamin C, causing different color changes and, then, influencing consumer preferences [52].

## 5. Conclusions

Cold plasma can efficiently inactivate *A. acidoterrestris* in both saline and apple juice. The input power, gas flow rate, and treatment time of cold plasma significantly influence the inactivation. Directly cold plasma-produced singlet oxygen dominates the inactivation of *A. acidoterrestris* at the voltage of 1.32 kV. The membrane integrity and the contents of DNA, proteins, lipids, and carbohydrates of the bacteria are destroyed by cold plasma. A low voltage (1.32 kV) can efficiently inactivate *A. acidoterrestris* within 60 s and only has a slight influence on the whole apple juice. More research needs to be conducted on how to prevent the bacteria from entering the VBNC state to guarantee the effectiveness of inactivation. Furthermore, more research on cold plasma processing parameter optimization or cold plasma in combination with other methods should be performed in the future to reduce the impact on apple juice quality and achieve a good inactivation outcome.

## Figures and Tables

**Figure 1 foods-12-01531-f001:**
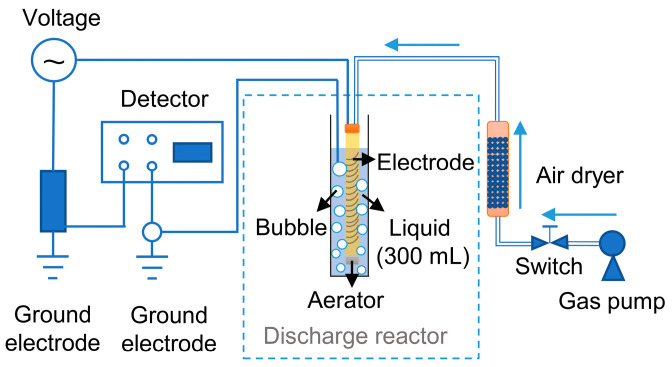
Schematic of the experimental setup of cold plasma device.

**Figure 2 foods-12-01531-f002:**
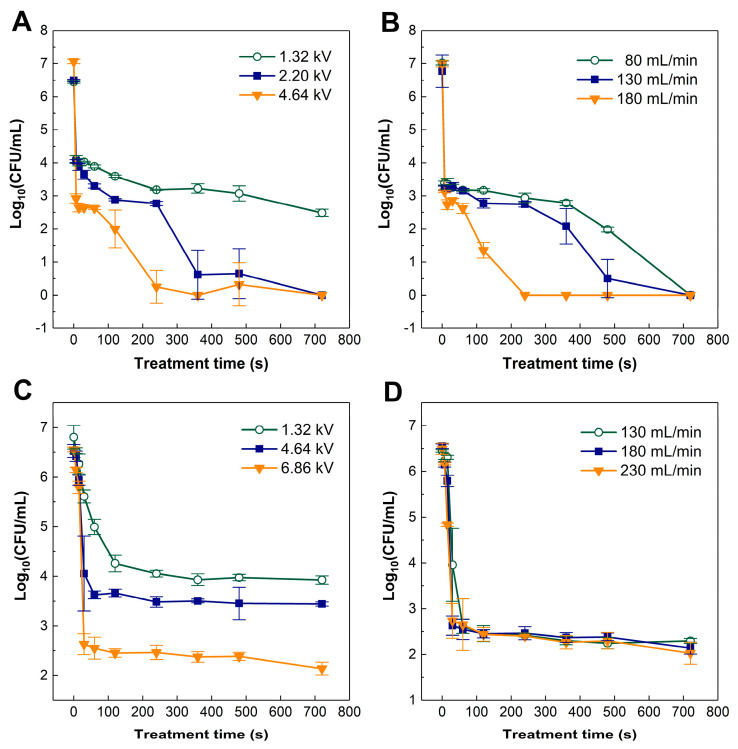
Survival curves of *A. acidoterrestris* in 0.85% saline (**A**,**B**) and 12% apple juice (**C**,**D**) treated by cold plasma with different input powers (**A**,**C**) and gas flow rates (**B**,**D**). The mean value and standard deviation calculated from three replicates are shown by each point and error bar, respectively.

**Figure 3 foods-12-01531-f003:**
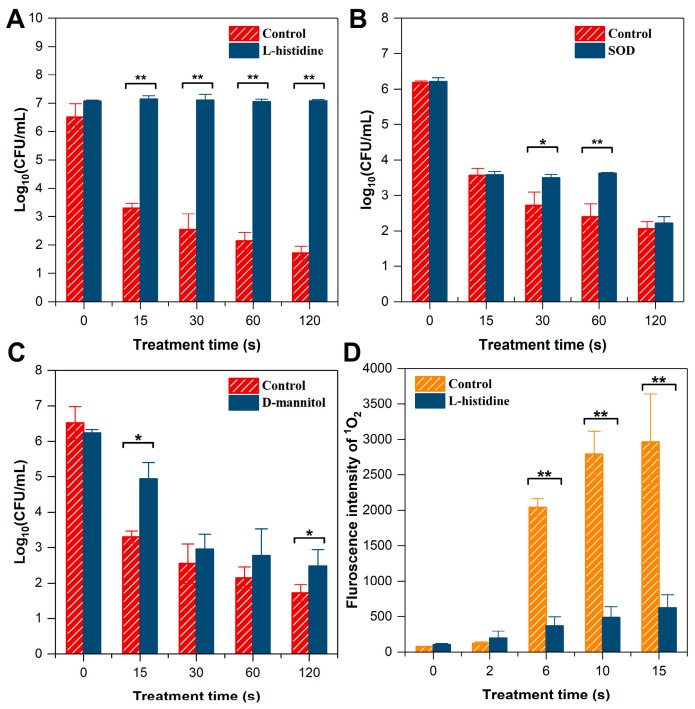
Experiments for scavenging singlet oxygen (^1^O_2_), superoxide anion (•O_2_^−^), and hydroxyl radical (•OH) by 1 M L-histidine (**A**), 500 U SOD (**B**), and 1 M D-mannitol (**C**), respectively. The positive experiments for validating the concentration of singlet oxygen with SOSG (10 μM) and with or without 1 M L-histidine are shown in (**D**). The cold plasma voltage was set at 1.32 kV and the gas flow rate was set at 130 mL/min. All experiments were repeated in triplicate. One-way ANOVA was used for calculating statistical significance. *, *p* < 0.05; **, *p* < 0.01.

**Figure 4 foods-12-01531-f004:**
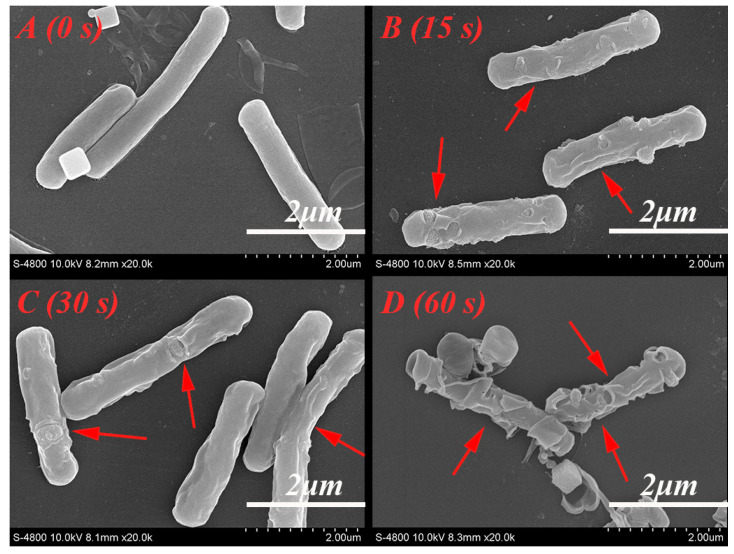
SEM images of (**A**) untreated *A. acidoterrestris* and (**B**–**D**) *A. acidoterrestris* treated by cold plasma for 15, 30 and 60 s, respectively, with a voltage of 1.32 kV and a gas flow rate of 130 mL/min. The red arrows indicate severe damage points on the cells. The magnification is ×20,000.

**Figure 5 foods-12-01531-f005:**
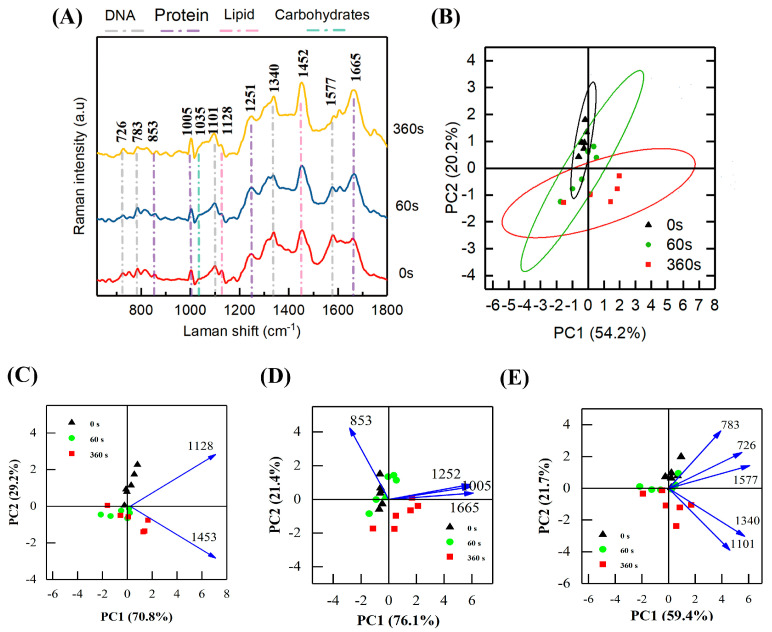
Raman spectra (**A**) and principal component analysis plot (**B** for all bands, **C** for bands of lipids, **D** for bands of proteins, and **E** for bands of DNA) of *A. acidoterrestris* treated by cold plasma (input power: 1.32 kV; gas flow: 130 mL/min) at 0, 60, and 360 s, respectively.

**Figure 6 foods-12-01531-f006:**
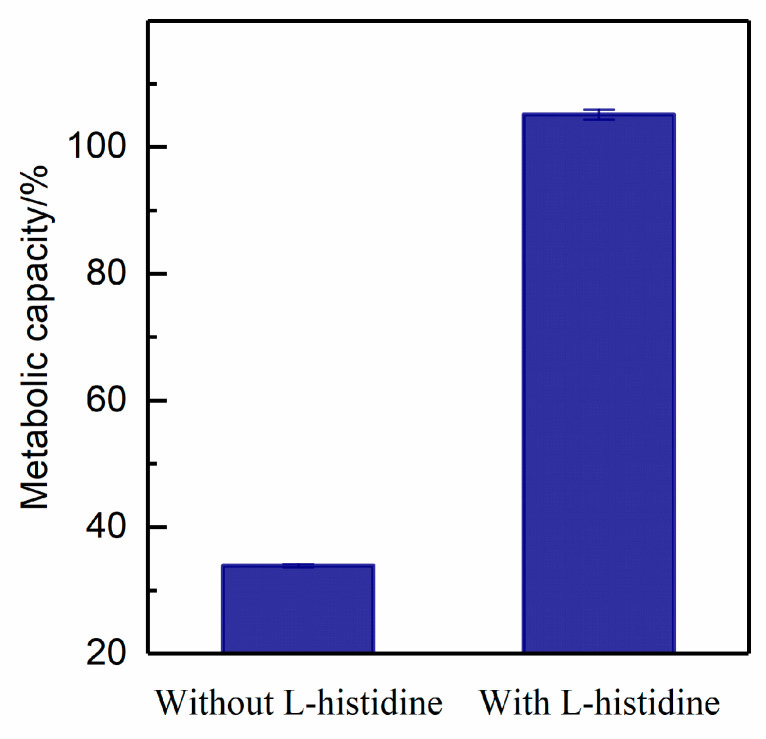
The percentage of *A. acidoterrestris* with metabolic capacity after being treated by cold plasma (input power: 1.32 kV; gas flow: 130 mL/min) for 60 s with and without L-histidine, as measured by a resazurin-based assay kit; Ex/Em: 560/590 nm. The investigation set the plasma-untreated sample as 100%.

**Figure 7 foods-12-01531-f007:**
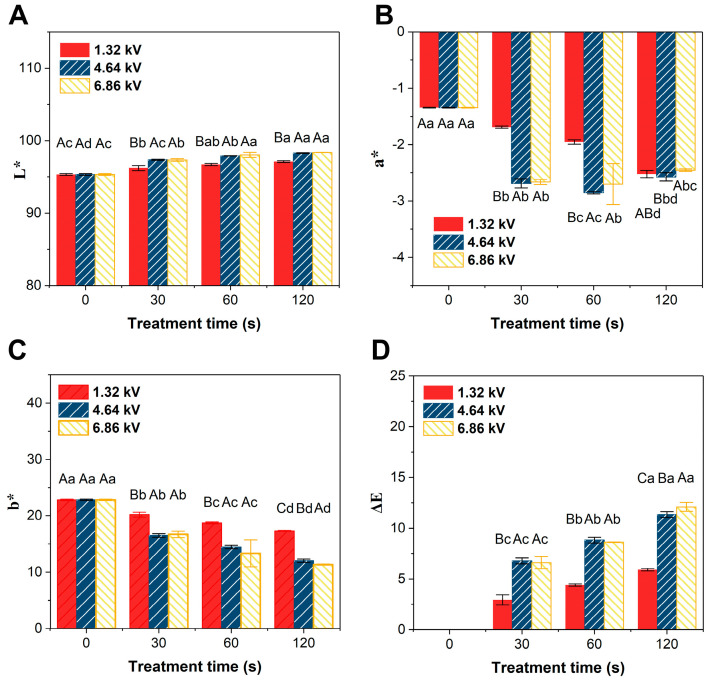
Effects of input power (1.32, 4.64, and 6.86 kV) and treatment time (0, 30, 60, and 120 s) on the color parameters of apple juice treated by cold plasma. (**A**) L* represents “brightness/darkness”, (**B**) a* represents “redness/greenness”, (**C**) b* represents “blueness/yellowness”, and (**D**) ΔE represents the total color difference. The experiments were repeated three times in duplication, and the results are expressed as mean ± standard deviation (SD). For treatments with different input power (uppercase) and treatment time (lowercase), the means with different letters demonstrate a statistically significant difference (*p* < 0.05) between them.

**Table 1 foods-12-01531-t001:** Kinetic parameters and evaluation of the biphasic model of the inactivation of *A. acidoterrestris* in apple juice.

Input Power (kV)	Kinetic Parameters				Modeling Efficiency	
	*f*	*k_max_* _1_	*k_max_* _2_	*lgN* _0_	RMSE	Adj-R^2^
1.32	0.99 ± 0.00	0.07 ± 0.01	0.00 ± 0.00	6.73 ± 0.07	0.11	0.99
4.64	0.99 ± 0.00	0.20 ± 0.50	0.00 ± 0.00	6.85 ± 0.20	0.25	0.96
6.86	1.00 ± 0.00	0.30 ± 0.05	0.00 ± 0.00	6.99 ± 0.33	0.42	0.95

**Table 2 foods-12-01531-t002:** Effects of input power (1.32, 4.64, and 6.86 kV) and treatment time (0, 30, 60, 120, and 360 s) on the physicochemical indicators of apple juice after cold plasma treatment.

Input Power/kV	Time/s	TSS/%	pH	TS (g/100 mL)	RS (g/100 mL)	TA (g/100 mL)	TP (g/100 mL)
1.32							
	0	12.13 ± 0.12 ^a^	3.74 ± 0.08 ^a^	10.89 ± 0.15 ^a^	9.36 ± 0.06 ^a^	0.24 ± 0.00 ^a^	0.17 ± 0.01 ^a^
	30	12.07 ± 0.12 ^a^	3.66 ± 0.01 ^b^	10.85 ± 0.51 ^a^	9.32 ± 0.19 ^a^	0.24 ± 0.01 ^a^	0.17 ± 0.00 ^b^
	60	12.02 ± 0.04 ^a^	3.69 ± 0.05 ^a^	10.92 ± 0.19 ^a^	9.23 ± 0.23 ^a^	0.23 ± 0.00 ^a^	0.16 ± 0.00 ^c^
	120	12.02 ± 0.04 ^a^	3.78 ± 0.06 ^a^	10.75 ± 0.24 ^a^	9.18 ± 0.20 ^a^	0.23 ± 0.00 ^a^	0.16 ± 0.00 ^c^
4.64							
	0	12.13 ± 0.12 ^a^	3.74 ± 0.08 ^a^	10.89 ± 0.15 ^ac^	9.36 ± 0.06 ^b^	0.24 ± 0.00 ^a^	0.17 ± 0.01 ^a^
	30	11.98 ± 0.04 ^a^	3.70 ± 0.06 ^a^	10.92 ± 0.49 ^ac^	9.23 ± 0.20 ^b^	0.23 ± 0.00 ^a^	0.16 ± 0.01 ^b^
	60	11.80 ± 0.12 ^a^	3.66 ± 0.02 ^a^	10.67 ± 0.22 ^bc^	9.71 ± 0.16 ^a^	0.24 ± 0.01 ^a^	0.14 ± 0.01 ^c^
	120	11.97 ± 0.05 ^a^	3.68 ± 0.06 ^a^	11.28 ± 0.07 ^a^	9.29 ± 0.15 ^b^	0.24 ± 0.00 ^a^	0.13 ± 0.00 ^c^
6.86							
	0	12.13 ± 0.12 ^a^	3.74 ± 0.08 ^a^	10.89 ± 0.15 ^a^	9.36 ± 0.06 ^a^	0.24 ± 0.00 ^a^	0.17 ± 0.01 ^a^
	30	12.02 ± 0.08 ^a^	3.68 ± 0.04 ^a^	10.97 ± 0.39 ^a^	9.32 ± 0.07 ^a^	0.24 ± 0.00 ^a^	0.15 ± 0.01 ^a^
	60	11.92 ± 0.04 ^a^	3.76 ± 0.03 ^a^	10.30 ± 0.41 ^b^	9.33 ± 0.15 ^a^	0.24 ± 0.00 ^a^	0.14 ± 0.00 ^b^
	120	11.92 ± 0.04 ^a^	3.74 ± 0.03 ^a^	10.73 ± 0.22 ^ab^	9.29 ± 0.20 ^a^	0.24 ± 0.00 ^a^	0.13 ± 0.00 ^c^

The experiments were repeated three times in duplication, and the results are expressed as mean ± standard deviation (SD). The means in the same column not sharing a common superscript letter differ significantly (*p* < 0.05). Abbreviations: TSS, total soluble solids; TS, total sugar; RS, reducing sugar; TA, titratable acidity; TP, total polyphenol.

## Data Availability

Data are contained within the article.

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
