# Peer review of "Role and Mechanism of Cold Plasma in Inactivating Alicyclobacillus acidoterrestris in Apple Juice"

_foods, 2023, doi:10.3390/foods12071531_

Round 1

Reviewer 1 Report

The paper is well organized, the objectives are clearly defined, the methodology is clear and the results are interesting. 

Congratulations and best regards.

Author Response

Dear reviewer,

Thank you very much for taking your precious time to review my paper and give valuable comments. I am very glad to receive your approval. 

Best wishes to you.

Yours sincelely,

Tianli Yue

Reviewer 2 Report

Section            Line     Comment

Title                 3          acidoterrestris in italics

Abstract          21        Check that the order or the experiments are the same through                                            the paper

M&M              90        Which gas was used?

                        113      Why a cocktail was used?

Section            2.3       5 respectively. A bit confusing

Line                 125      Space before the reference

                        126      Missing a .

                        133      What is (12)?

                        148      Space before 15

                        151      Space after 485

                        194      What is (1)?

                        229      Error! What that means?

                        229      Please mention Table 1 before discussing kinetic data

                        236      Comment: Adj-R2 for 4.64 is closer to 6.86 than to 1.32?

                        262      Space after A.

                        265      Fix the features in the parentheses

                        275      Error! What that means?

                        287      Why only 600-1800 cm-1. What about polyphenols and other                                             compounds? Vitamins C

                        291      Figure 5A X-axis title. Is it correct?

                        292      Be consistent with the way you redact the figure titles.

                        295      Explain 5A first then 5B. You could do a PCA for each group of                                             biological molecules (proteins, carbohydrates, etc.). This might                                           give you more information on the disrupt molecules group type

                       308      VBNC abbreviation should be stated here not in 309

                       324      Error! What that means?

                       444      There is another (12) in the document?

                       483      Error! What that means?

                       526      References: Homogenize the way journal names are presented.                                          Some information is missing, page number, etc.

 General comments:

The paper is very interesting, cold plasma, I believe has a great potential and future broad applications. The document needs to be homogenized in many ways. The tables are just presented but not mentioned within the text. English style, order of issues, figures titles, etc. An English proofing will be of great help.

Author Response

Dear reviewer,

We would like to thank you for your careful reading, helpful comments, and constructive suggestions, which has significantly improved the presentation of our manuscript.We have studied the valuable comments from you and tried our best to revise the manuscript. Please see the Response to Reviewer and  the revised manuscript in the attachment. Thanks again. Best wishes to you!

Yours sincerely,

Tianli Yue

Reviewer 3 Report

  1. I believe the study is intriguing, but to make it more comprehensive, it would be beneficial to measure the antioxidant capacity of apple juice both before and after treatment with cold plasma. It would be a great addition to the study if this section could be included to determine the changes in antioxidant capacity due to the cold plasma treatment.

  2. Based on the findings, the cold plasma treatment appears to alter the appearance of the apple juice compared to untreated samples. Would you suggest using this technique to inactivate Alicyclobacillus?

Author Response

Dear reviewer, 

 We would like to thank you for your careful reading, helpful comments, and constructive suggestions, which has significantly improved the presentation of our manuscript. 

Point 1: I believe the study is intriguing, but to make it more comprehensive, it would be beneficial to measure the antioxidant capacity of apple juice both before and after treatment with cold plasma. It would be a great addition to the study if this section could be included to determine the changes in antioxidant capacity due to the cold plasma treatment.

Response 1: We sincerely appreciate the reviewer's careful reading and insightful comments. We agreed with your recommendations that measuring apple juice's antioxidant capacity would be a great addition to the study. The experimenter, however, has graduated from our school with no apple juice sample remaining. We feel sorry that the experiment can not be conducted. But thanks for the good advice, we will test this indicator in a future study.

Point 2: Based on the findings, the cold plasma treatment appears to alter the appearance of the apple juice compared to untreated samples. Would you suggest using this technique to inactivate Alicyclobacillus?

Response 2: We sincerely thank the reviewer for careful reading and good questions. We think the influence on apple juice caused by short-time cold plasma treatment with low voltage is within limits of acceptability. However, we suggest cold plasma combined with other sterilization techniques to be used on inactivating Alicyclobacillus, for better sterilization with preserving the quality of juice.

Thank you so much. Best wishes to you.

Yours sincerely,   

Tianli Yue